# Mental health and mental well-being of Black students at UK universities: a review and thematic synthesis

Nkasi Stoll ,[1] Yannick Yalipende,[2] Nicola C Byrom,[3] Stephani L Hatch,[1,4] Heidi Lempp[5]

[1]Department of Psychological Medicine, King's College London Institute of Psychiatry Psychology and Neuroscience, London, UK
[2]Division of Psychology and Language Sciences, University College London, London, UK
[3]Department of Psychology, King's College London Institute of Psychiatry Psychology and Neuroscience, London, UK
[4]ESRC Centre for Society and Mental Health, King's College London, London, UK
[5]Department of Inflammation Biology, King's College London Faculty of Life Sciences and Medicine, London, UK

**Correspondence to**
Ms Nkasi Stoll;
nkasi.1.stoll@kcl.ac.uk

## ABSTRACT

**Background** There is a knowledge gap about the experiences that affect the mental health of Black university students in the UK. Current research is focused on understanding the continuation, attainment and progression gap between Black students and non-Black students. It is essential to know more about the interactions between personal and institutional factors on the mental health of Black students to explain the inequalities in their experiences and outcomes across the university lifecycle. The current study set out to thematically synthesise articles that explore the experiences that affect the mental health and mental well-being of Black university students in the UK.

**Methods** This study is a qualitative thematic synthesis of a literature review. We developed search strategies for four online databases (PubMed, Social Science Premium Collection via ProQuest, Open Access Theses and Dissertations, and Open Grey) covering January 2010 to July 2020. This search was combined with a manual search of reference lists and related citations. All articles in English addressing mental health and mental well-being experiences among Black university students studying at a UK university were included. Critical Appraisal Skills Programme Checklist was used to assess bias. A thematic synthesis was conducted using Braun and Clarke (2006)'s six-step guide to develop descriptive themes and analytical constructs.

**Results** Twelve articles were included. Several themes were identified as affecting the mental health of Black university students in the UK: academic pressure, learning environment, Black gendered experience, isolation and alienation, culture shock, racism and support.

**Discussion** This review provides an appraisal of the factors affecting the mental health and mental well-being of Black students at UK universities, which need to be addressed by higher education policy-makers and key decision-makers. Further research is needed about the mental health experiences of Black university students in relation to Black identities, suicidality, mental health language, the physical environment, and racism and other institutional factors.

## INTRODUCTION

There is increasing concern about the mental health and mental well-being of university students in the UK.[1–4] The university student

### Strengths and limitations of this study

► This is the first review and thematic synthesis to provide an overview of the personal and institutional experiences that affect the mental health and mental well-being of Black students studying at UK universities.
► A thematic synthesis approach was used to describe and compare the main findings, to help inform policy and interventions.
► Database and manual searching were extensive, screening 369 articles.
► It is possible not all relevant articles were found due to inconsistent terminology for the racial category 'Black' and 'mixed' in existing studies.

lifecycle has been conceptualised in terms of (1) the application experience which covers the interactions between potential students and the institution up to the point of arrival; (2) the academic experience which encompasses students' interactions with their institution of study; (3) the campus experience, which includes students' life not directly connected with study and may include activities away from the campus and (4) the graduate experience which covers the institution's role in assisting students' transition to employment or further study.[5] Throughout the lifecycle,[5] British and international students are exposed to a range of experiences that make universities a high-risk period for maladaptive coping and the possible onset of poor mental health. These experiences include (but are not limited to): individuation, separation from family, new social connections, increased autonomy and responsibility, academic-related stress, financial concerns, sleep disruption, balancing conflicting demands of studying with personal and family life, and exposure to risky behaviours, including recreational drug use and alcohol binging.[6–10] Late adolescence through young adulthood is a peak age period

for the first onset of mental health problems,[11] which overlaps the period that many young adults enter and navigate the university student lifecycle.[5] Unsupported mental health problems are associated with progression to other comorbid disorders, substance use disorders, self-harm and suicide ideation and attempts.[12–15] According to Higher Education Statistics Agency, there are 2.4 million university students in higher education in the UK,[16] which means that universities are in the best position to provide prevention and intervention to many young adults during a critical transitional period. In the UK, Black students are more likely than white students to self-report high levels of engagement and participation in their university studies; covering seven categories of engagement including course challenge, critical thinking and research and inquiry.[17] Despite high engagement levels, Black students report lower satisfaction and are less likely to complete their course, achieve a first-class (70% or above) or upper second-class (60% or above) UK degree classification and progress to further education.[18] In addition to the aforementioned experiences and risk factors for poor mental health, there is tentative research around the institutional issues affecting Black university student mental health including inequality in access, under-representation, the attainment gap, institutional racism and racist curriculum.[19–23] To explain the inequalities in student experiences and outcomes across the university lifecycle, further understanding of the interactions between personal and institutional factors on the mental health of Black students' is needed. This review aims to synthesise the results of existing studies that explore mental health and mental well-being in Black students at UK universities. Therefore, our research question is as follows: what are the experiences that affect the mental health and mental well-being of Black university students in the UK, as reported in the literature?

## METHODS
### Design
A review of the literature including quantitative, qualitative and mixed studies was conducted on peer-reviewed and non-peer-reviewed articles, addressing the mental health and mental well-being of Black students at UK universities. Reporting of this review was guided by the Enhancing Transparency of Reporting the Synthesis of Qualitative Research framework,[24] as the included articles were all qualitative.

### Search methods
The following databases were searched for the period of January 2010 to July 2020: PubMed, Social Science Premium Collection via ProQuest, Open Access Theses and Dissertations and Open Grey. The search strategy included several combinations of the following medical subject heading (MeSH) terms and keywords:[25] "mental health" or "psychological distress" or "mental wellbeing" and "university student" and "Black Caribbean" or "Black African" or "Black Mixed" or "Black Ethnic*".

### Data collection
All search results were saved to Zotero bibliographic management software and duplicates were removed. To be included in the review, articles had to (1) focus on the mental health or mental well-being of Black students at UK universities (2) apply empirical (eg, quantitative, qualitative or mixed-method research) or non-empirical research methods (eg, theoretical, opinion-based, student transcripts, policy papers) (3) be written in English, (4) be conducted in the UK. The first author (NS) removed all articles that did not fit the inclusion criteria after full-text review. NS manually searched reference lists of eligible articles and contacted lead authors to search for unpublished literature. The authors decided to be lenient in the inclusion criteria during the screening and inclusion of articles because existing research on Black university student mental health is limited, and to enable the inclusion of articles that add value to the discussion of the topic. All included articles were critically appraised by NS and the second author YY using the 'Critical Appraisal Skills Programme' (2018)[26] checklist for qualitative research.

### Data analysis and synthesis
For each article, all text from 'Results/Findings' and 'Discussion' were extracted and imported into NVivo V.12 software (NVivo Qualitative data analysis Software; QSR International, V.12, 2018). Study characteristics were extracted into a Microsoft Excel grid by NS and revised by YY. Full-text screening, selection, data extraction and critical appraisal of all articles was conducted prior to data analysis. A thematic analysis was conducted to describe and compare the main findings, following Braun and Clarke's[27] six-step guide. NS carefully read each article to familiarise themselves with the content, annotating initial ideas for codes and themes. NS then read and reread the articles to develop an initial coding scheme, using a constant comparative approach.[28] Themes and subthemes were identified to capture important patterns across and within the articles included. Six articles (50% of the sample) were randomly selected to be coded independently by YY, to provide validation. The codes were examined for similarities and differences and then organised into a hierarchy to create the final codes. Any differences were resolved by SH. The final codes were validated by NS, YY and YY, before being synthesised into the final thematic framework presented in this paper to minimise researcher subjectivity and improve the credibility of the work.

### Patient and public involvement
The lead author is a Black Caribbean PhD student, and the second author is a Black African master's student. Both were involved in the design, conduct, reporting and dissemination of this research.

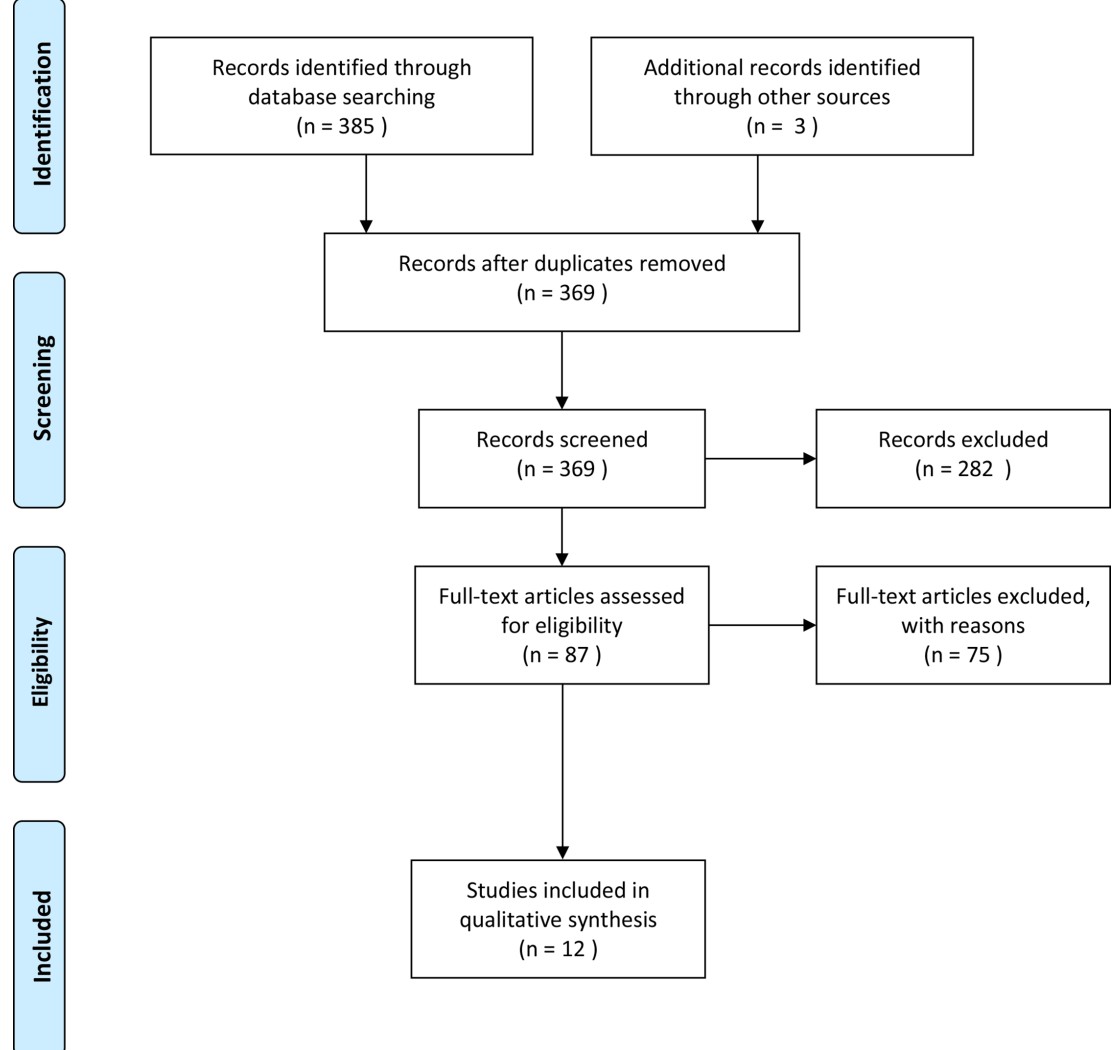

**Figure 1** PRISMA flow diagram. PRISMA, Preferred Reporting Items for Systematic Reviews and Meta-Analyses.

## RESULTS

### Included studies

Figure 1 shows the number of articles identified and rejected at each stage of the review process.[29] The search strategies identified 388 references of which 369 were kept after removing duplicates. After screening, 87 articles were read completely. Of these, 75 articles were excluded. A final sample of n=12 studies were included. Table 1 presents the characteristics of the included studies.

All included articles were assessed by NS and YY as reasonable quality and were therefore judged to have relevant contributions for the thematic synthesis (table 2).

### Synthesised findings

The concept matrix for the findings can be seen in table 3. We interpreted seven analytical themes derived from the participants' beliefs, perceptions and experiences, and the article's authors' interpretations. The references for articles contributing to each of the themes are provided.

### Academic pressure

In 6 out of 12 articles, academic pressure was a determinant of mental distress for Black students.[30] Students described their mental health as being adversely affected by the multiple and simultaneous academic tasks they were expected to complete during their degree.[31] Cumulatively, Black students reported their observation that, unlike their non-Black peers, they faced pressures from their families and the wider Black community which meant they needed to work harder to succeed at university; which further exacerbated mental distress.[31–33] This pressure came from their families continuously reminding them of the sacrifices and investments made to support their education, specifically immigration and financial strain.[30 31] Black students with pre-existing mental and physical health conditions have been reported to be especially vulnerable to poor mental health caused by this academic pressure.[31 34] There was some evidence that Black students might not seek mental health support and instead prioritise their continuation and success in academia, at the expense of their mental health, as they

**Table 1** Characteristics of included studies

| Author(s) | Database | Type of article | No of Black participants | Subject characteristics | Location | Psychological phenomena explored | Methods |
|---|---|---|---|---|---|---|---|
| Hayford | Hand searched | Dissertation | 6 | One was Caribbean, one was Somalian, one was Nigerian, two were mixed Black African, and one was Ghanaian | East Anglia | Well-being | Interviews |
| Jones[31] | PubMed | Blog | 1 | Black academic staff | UK | Well-being | Personal reflections and notes |
| Alloh et al[34] | Proquest | Journal article | 9 | International Nigerian students studying at the masters level in a UK university | Southeast of England | Well-being | Interviews |
| Arday[32] | Proquest | Journal article | 14 | 32 BME university students | UK | Mental Health | Interviews focus Groups |
| Jackson-Cole[33] | Open Grey | Doctoral thesis | 10 | Postgraduate students. two Black or Black British – Caribbean participants, seven Black or Black British –African, one Mixed –white and Black African | England | Well-being | Interviews |
| Myrie and Gannon[30] | Hand searched | Journal article | 3 | 2 Black African students, 1 Black African and Caribbean student | East and North London | Well-being | Interviews |
| Bunce et al[35] | PubMed | Journal article | 14 | 14 full-time students from two degree programmes in health and social care related subjects—Black African (12) and white and Black Caribbean (2) | Unknown | Well-being | Focus groups |
| Cummings[36] | Open Grey | Doctoral thesis | 5 | Black women with Caribbean heritage, who were being educated at a university in the East Midlands | East Midlands | Well-being | Focus groups |
| Akel[37] | PubMed | Report | Not specified | 195 BME students. 'Black' refers to people of African heritage and the diaspora. | London | Well-being | Interviews focus groups |
| Bhatti[38] | Open Grey | Doctoral thesis | 5 | Black students | London | Body image | Focus groups |
| Crittle[39] | Proquest | Blog | 1 | African American student | UK | Stress | Personal reflections and notes |
| Baxe[40] | PubMed | Conference notes | 1 | Black students | UK | Mental Health | Personal reflections and notes |

**Table 2** Critical Appraisal Skills Programme Checklist for included articles

| | Was there a clear statement of the aims of the research? | Is a qualitative methodology appropriate? | Was the research design appropriate to address the aims of the research? | Was the recruitment strategy appropriate to the aims of the research? | Was the data collected in a way that addressed the research issue? | Has the relationship between researcher and participants been adequately considered? | Have ethical issues been taken into consideration? | Was the data analysis sufficiently rigorous? | Is there a clear statement of findings? | How valuable is the research? |
|---|---|---|---|---|---|---|---|---|---|---|
| Hayford | Yes | Yes | Yes | Yes | Yes | Yes | Yes | Yes | Yes | Valuable |
| Jones[31] | No | N/A | N/A | N/A | N/A | N/A | N/A | N/A | Yes | Valuable |
| Alloh et al[34] | Yes | Yes | Yes | Yes | Yes | No | Yes | Yes | Yes | Valuable |
| Arday[32] | Yes | Yes | Yes | Yes | Yes | No | Can't Tell | Yes | Yes | Valuable |
| Jackson-Cole[33] | Yes | Yes | Yes | Yes | Yes | Yes | Yes | Yes | Yes | Valuable |
| Myrie and Gannon[30] | Yes | Yes | Yes | Yes | Yes | No | Yes | Yes | Yes | Valuable |
| Bunce et al[35] | Yes | Yes | Yes | Yes | Yes | No | Can't Tell | Yes | Yes | Valuable |
| Cummings[36] | Yes | Yes | Yes | Yes | Yes | Yes | Yes | Yes | Yes | Valuable |
| Akel[37] | Yes | Yes | Yes | Yes | Yes | No | Yes | Yes | Yes | Valuable |
| Bhatti[38] | Yes | Yes | Yes | Yes | Yes | Yes | Yes | Yes | Yes | Valuable |
| Crittle[39] | Yes | Yes | Yes | Yes | Yes | Yes | Yes | Yes | Yes | Valuable |
| Baxe[40] | No | N/A | N/A | N/A | N/A | N/A | N/A | N/A | Yes | Valuable |

N/A, not available.

**Table 3** Concept matrix for included studies

| | Support | Isolation and alienation | Racism | Culture shock | Black gendered experience | Learning environment | Academic pressure |
|---|---|---|---|---|---|---|---|
| Hayford | x | x | | x | | | |
| Jones[31] | x | | x | x | | | x |
| Alloh et al[34] | x | | x | x | | | x |
| Arday[32] | x | x | x | | x | | x |
| Jackson-Cole[33] | x | | x | | x | x | |
| Myrie and Gannon[30] | x | x | | | x | | |
| Bunce et al[35] | | x | | x | | x | |
| Cummings[36] | x | x | x | | x | x | |
| Akel[37] | | x | x | | | x | |
| Bhatti[38] | x | | | | x | | |
| Crittle[39] | x | x | | | | | |
| Baxe[40] | | x | | | | | |

feared a mental illness diagnosis might affect their success in academia.[32]

## Learning environment

In 4 of the 12 articles, researchers described Black students' perceptions of the university learning and teaching environment and how it impacted their mental health and mental well-being.[33 35–37] Lectures that had discussions on diversity, inclusion, ethnicity, race, and identity were said to be cathartic and liberating for Black students, which contributed to positive well-being.[35–37] Black students reported they had to censor themselves in academic spaces to be seen by white students and staff as acceptable and agreeable instead of loud, disruptive and confrontational.[36] This behaviour was believed to improve their learning experience and relationship with teaching staff, at the expense of their well-being. Participants spoke about their lack of relatedness to the white majority academic staff, students and teaching materials; and racism in the classroom made them feel excluded, frustrated, distressed, discouraged and unmotivated to engage in their degree course.[33 35–37] In their university seminars, Black students recalled Black people were racially stereotyped as being bad people, criminals and having lower intellectual ability by white students, and these stereotypes were sometimes reinforced by course materials.[35–37] Black female-identifying students recounted their academic knowledge was overpoliced and overscrutinised by teaching staff, and inaccessible learning and well-being support compared with men and white women on their course, which impacted their mental health and learning experience.[33]

## Black gendered experience

Black students' experiences of multiple marginalisation in the forms of racism and sexism was discussed in five articles.[30 32 33 36 38] Black male-identifying students reported being affected by the discourses of hypermasculinity,

which put pressure on them to not show their emotions, cope with their mental distress alone, and continue with their studies on their own.[30 32] This caused them to feel shame when seeking mental health support. For some female-identifying Black students, feelings about their health, ideal body shape, and size were influenced by the beliefs, culture, race, and ethnicity of the people in their lives.[38] The importance of seeing their body shape among Black students who looked like them was discussed.[38] Some students felt the white body ideal is not the same as the Black body ideal, therefore, visiting their country of origin positively impacted their stress levels, self-esteem and body image. Social media and the music industry were mentioned as vehicles in which the ideal Black female body is narrated, scrutinised and picked apart.[38]

## Isolation and alienation

Eight out of 12 included studies discussed isolation and alienation.[30 32 35–37 39 40] Black students who struggled with their mental health experienced stigma from their communities (including friends and family) before attending university, leaving them isolated from those communities (A Hayford, unpublished data, 2019). At university, participants reported being ignored or avoided by their non-Black peers in learning, social and living environments which led to feelings of lack of belonging, diminished their overall well-being, and evoked a range of negative emotions, including discomfort, distress, frustration and anger.[32 35–37 39 40] Those with little or no mental health support from family felt isolated and alienated from their friends, family, peers and academic staff, and used denial to cope with the lack of support, which exacerbated their mental health problems.[32]

Black students reported that a lack of relatedness to white students affected their well-being, undermined their motivation for academic success, and had a negative influence on their sense of autonomy and competence.[35–37] Black

students mentioned white students did not understand that they have to reconstruct their identity in the white image and practice silence to be successful at university and that this process negatively affected their mental health (A Hayford, unpublished data, 2019).

Participants in one article discussed that academic staff do not always create safe and protective learning and living spaces for Black university students.[37] As a result, Black students reported they had to defend themselves by verbally challenging prejudicial beliefs shared in classrooms, which made white students uncomfortable, leaving Black students feeling bullied, alienated and alone. Participants who had the support of other Black students in their learning environment at university reported being more comfortable, understood and happy.[35] However, this may not apply to Black students who are international students. One article reported a student being in tears, feeling alone and isolated, and almost dropping out of university due to their African accent isolating them from Black British students and non-Black students.[35]

### Culture shock

There was evidence suggesting Black international-status students had to adjust to new environments, cultures, and lifestyles in the UK which was characterised by changes in diet, weight, physical activity, sleep, alcohol and smoking consumption, and led to psychological stress, depression, loneliness and loss of confidence.[34 35] Black students who emigrated from a collectivist culture to the UK experienced culture shock during the transition from home to their new environment, that was characterised by worry, stress and insomnia.[34] Contrastingly, there was evidence[34] to suggest Black home-status students had slightly poorer mental health and lower self-esteem than Black international-status students. Indeed, one article reported some Black international-status students believed their stress levels to be higher in their home country which gave them the perspective of living in the UK as being less stressful.[34]

### Racism

Five of the 12 included studies explored how institutional racism, discrimination and hegemonic white privilege made universities toxic spaces for Black students, which affected their mental health and well-being.[31–33 36 37] The constant stress of being confronted with racism (including racial microaggressions), discrimination, and having to survive hostile racist environments at university led to poor mental health and mental illness in Black students.[31–33 36] Racism had negative consequences on Black students' sense of belonging at university, motivation to socialise, their interactions with white students and staff, and their academic achievements and progression, which led to further mental distress. There was evidence to suggest Black students' recognition of racial difference had a greater impact on their mental well-being and potential or experienced mental health difficulties, than evidence that others treated them differently on account of their race.[36] Participants felt unable to express their feelings of racism and difference from white peers to others for fear they would not be understood, would be judged or rejected by peers and university staff, or due to uncertainty that these feelings were valid or legitimate.[36] These unarticulated feelings resulted in disengagement from university study and services.

### Support

The aforementioned factors that possibly influence the mental health of Black students were discussed to affect and be affected by the type, access to, experience and perceptions of mental health support in 9 of the 12 included studies (A Hayford, unpublished data, 2019).[30 36–39]

Black students reported they had to take care of their mental health by trying to remain positive and use distraction, use psychoeducational resources, change daily habits including exercise and nutrition, and use food to cope with stress at university and connect them to family.[32–34 36 38]

Access to mental health support from family, friends and peers was identified as a contributory factor to positive well-being and success at university (A Hayford, unpublished data, 2019).[31–33 36 39] One article reported Black students who did not have familial or social capital found accessing university mental health support more difficult.[32] Four included articles[30–32 39] found mental health stigma left Black students feeling unsupported by their family and unable to get the professional support needed. Black-only spaces at university were reported as important to obtain support for mental health and well-being (A Hayford, unpublished data, 2019).[31 32 36 39] These spaces gave students comfort, where they were not judged and misunderstood by non-Black students and staff; and could be with students from a shared background and experience. These Black-only spaces were especially important when there was a lack of Black students in classroom spaces, and could be online or in person. Black students also placed importance on mental health support from an understanding religious organisation or chaplain at their university (A Hayford, unpublished data, 2019).[31 32]

Included articles described Black students' beliefs that university services failed to provide appropriate support because they were institutionally racist and treated them like a monolithic group; and did not take responsibility for failing Black students (A Hayford, unpublished data, 2019).[31 32 37] Participants' awareness that white mental health practitioners at university services were the majority race prevented them from accessing services, because of perceived racial stereotyping and lack of trust.[32 37] Some Black students avoided university mental health services to such an extent that they were not knowledgeable about what was available for them (A Hayford, unpublished data, 2019). Black students who needed mental health support did not seek such services until in crisis (A Hayford, unpublished data, 2019).[32] and when they did, faced long waiting times.[32] Black students

who did receive help from university mental health services reported experiencing racialised stereotypes that affected their relationship with white mental health practitioners (A Hayford, unpublished data, 2019).[31 32 37] They had to educate practitioners on Black culture and the Black experience (which included how racism and microaggressions affected their mental health), which was a burden on their mental health and well-being (A Hayford, unpublished data, 2019).[31 32 37] Because of this, students felt more comfortable with a Black, Asian or other minoritised ethnic mental health practitioner (A Hayford, unpublished data, 2019).[32 37] Black students described a sense of helplessness regarding their mental healthcare treatment (A Hayford, unpublished data, 2019).[32 37] One article discussed Black students felt unable to challenge mental health professionals about their healthcare treatment because they may be labelled as difficult or unstable, and worsen their relationship with the mental healthcare provider.[32]

## DISCUSSION

This review and thematic synthesis provides evidence on how the following experiences affect the mental health and mental well-being of Black university students in the UK: academic pressure, learning environment, Black gendered experience, isolation and alienation, culture shock, racism and support.

The study findings highlight that for Black university students their mental health and mental well-being may be deeply rooted in institutional factors or dimensions, largely racism,[31–33 36–38] which negatively affects most aspects of their higher education experiences, particularly in the learning, social and living environments. Critical Race Theory in Education (CRT-E) is used to engage with and work against racism in the context of UK higher education.[23 41 42] CRT-E theorists have explored racial inequality in admissions, curriculum and pedagogy, teaching and learning, institutional culture, campus racial climate, and policy and finance.[41–48] However, CRT-E has failed to interrogate the role mental health plays in racial inequality. The Office for Students[2] stated that 'Black students with mental health conditions are being failed throughout the student cycle' (pg. 6), evidenced by their data which shows Black full-time students who report a mental health condition have some of the lowest attainment, continuation and progression rates.[2] To understand and begin to address these concerns, further research is required into the role historically anti-Black racist systems embedded within UK universities plays on the mental health experiences of, risk factors for and challenges faced by Black university students affecting their mental health, experiences, outcomes and progression.

This review is very timely given the resurgence of the global Black Lives Matter movement and protests following the murder of George Floyd.[49] Growing conversations around the role race and institutional racism play on the mental health of Black people motivated students and staff to write multiple open letters[50–55] demanding UK universities do more to tackle racism, be antiracist and diversify and decolonise their curricula and institutions. Students and staff are calling out for UK universities to provide racially and culturally appropriate mental health support for their Black students. However, to achieve this, further research is needed to interrogate the Eurocentric, ethnocentric and egocentric ways in which institutional policy, research and mental health practices disadvantage different Black students from varying identities and social statuses across the university student lifecycle.

The literature largely demonstrates an interest in Black university student mental health. However, this is limited exploration into whether and how mental health experiences differ across different Black student identities (including gender, culture, ethnicity, nationality), social statuses and the impact of migration on student mental health. This is important given the finding that Black students believe universities treat them as a monolithic group (A Hayford, unpublished data, 2019).[30 32 37] There is a knowledge gap on suicidality among Black students. A recent national study[56] found the risk of suicide is lower among Black university students compared with white students (risk ratio 0.53 (95% CI 0.32 to 0.88)). This finding needs to be investigated further. One article included in this review (A Hayford, unpublished data, 2019) discussed that mental health is either not spoken about or spoken about differently in the Black community compared with within the wider UK university community. More studies are needed on mental health language and literacy.

The results of this review can assist academic staff, researchers, funders and policy-makers to identify potential areas of priority in student mental health services design, development and reform with Black students. Based on the review findings, the authors recommend further investigation into the following potential strategies: (1) in-person or virtual Black student mental health peer support groups; (2) decolonisation of mental health support services; (3) race, racism and mental health training for academic staff and mental health practitioners; (4) decolonisation of the classroom environment and teaching and learning materials; (5) scholarships and support funds for Black students;[57] (6) involvement of family in mental health support; (7) improve knowledge and accessibility of available services; (8) train and employ Black mental health professionals; (9) involve religious organisations and chaplains in mental health support and (10) targeted mental health support for international-status students.

### Strengths and limitations

To our knowledge, this is the first review and thematic synthesis to summarise the available literature on the experiences that affect the mental health and mental well-being of Black students studying at UK universities. By summarising these results, we make this information more readily available to students, academic staff,

policy-makers and researchers. An exhaustive literature search was conducted using multiple databases, and manual searching. Dissertations, blogs and unpublished articles were included alongside published articles. Our analysis allowed us to extract multiple recurrent themes pertaining to Black students' mental health. The results focus on both the personal and institutional factors that influence the mental health of Black university students.

The review has the following limitations. Despite the study selection criteria, no quantitative studies were included as they did not report the mental health experiences of Black university students. Second, although database and manual searching were extensive, it is possible relevant articles were missed due to inconsistent terminology for mental health and mental well-being and the racial category 'Black'. Third, the authors were unable to fully capture Black students' experiences as the citations selected from the included articles are a sample of the participants' subjective experiences and the researchers' interpretations. Fourth, articles that included Black, Asian and minoritised ethnic participants did not always clearly differentiate citations and themes derived specifically from Black student responses. The mixed racial category was not clearly defined in some articles so students who were of mixed Black heritage may have been missed in the analysis. Responses had to be drawn from participants' accounts disclosing their race. Finally, our analysis was limited to the 'results' and 'findings' sections of the selected articles; however, all sections of the articles were read to provide a deep understanding of the topic. We acknowledge that themes developed in this synthesis may be similar to findings reported in the included articles since it was not our intention to develop new interpretations or theories, and we wanted to stay close to the experiences reported by the participants in the included articles.

## CONCLUSION

This review and thematic synthesis offer an overview of the experiences that affect the mental health and mental well-being of Black students studying at universities in the UK as reported in the available literature. Our results suggest that there are both personal and institutional factors that affect the mental health and mental well-being of Black students. Further research into how the learning, social, living, cultural and physical environment of universities affects the mental health of Black university students; and the differences in the lived experiences of Black university students across different identities and social statuses would give valuable insights. The impact of institutional issues of racism and sexism on the mental health of Black students studying at UK higher education needs to be further researched and explicitly addressed by policy-makers and key decision-makers within universities.

**Acknowledgements** The authors would like to acknowledge Dionne Laporte for proof-reading the draft of this manuscript. NS carried out this work as part of a London Interdisciplinary Social Science Doctoral Training Partnership funded doctoral project.

**Contributors** NS (lead author and guarantor) conceived the study which was further refined by SLH (joint senior author) and HL (joint senior author). NS developed the detailed methodology, undertook database searches and title and abstract review. Each study retained for full-text review was reviewed by NS. YY reviewed 50% of the full text. Discrepancies between NS and YY regarding inclusion or exclusion were resolved by SLH. NS and YY made the critical appraisal of included studies. The thematic synthesis was conducted by NS and revised by YY, SLH and HL. All authors (NS, SLH, HL, YY and NCB) reviewed, commented and edited the manuscript and approved the final version.

**Funding** NS was supported by Economic and Social Research Council (grant number ES/P000703/1) via the London Interdisciplinary Social Science Doctoral Training Partnership. SLH is partly supported by the Economic and Social Research Council Centre for Society and Mental Health at King's College London (ES/S012567/1) and by the National Institute for Health Research (NIHR) Maudsley Biomedical Research Centre at South London and Maudsley NHS Foundation Trust and King's College London (BRC-1215-20018). HL currently receives funding for successful grants as a PI or co- PI: Wellcome Trust Institutional Strategic Support Fund (N/A); King's Health Partner Multiple Long-Term Conditions Challenge Fund (N/A); National Axial Spondyloarthritis Society (N/A); UKRCI Medical Research Council (MR/S001255/1); Medical Research Council (MR/R023697/1); National Institute for Health Research (RP-PG-0610-10066); Guy's and St. Thomas' Charity London (EFT151101); and vs Arthritis (N/A). NCB is partly supported by the Economic and Social Research Council funding for SMaRteN (ES/S00324X/1).

**Disclaimer** The views expressed are those of the authors and not necessarily those of the NHS, the NIHR or the Department of Health and Social Care, Wellcome Trust, ESRC or King's College London.

**Competing interests** None declared.

**Patient and public involvement** Patients and/or the public were involved in the design, or conduct, or reporting, or dissemination plans of this research. Refer to the Methods section for further details.

**Patient consent for publication** Not applicable.

**Ethics approval** This study does not involve human participants.

**Provenance and peer review** Not commissioned; externally peer reviewed.

**Data availability statement** No data are available.

**ORCID iD**
Nkasi Stoll http://orcid.org/0000-0003-0427-3367

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
