## [Reviewer comments · BMJ Open]

ARTICLE DETAILS

TITLE (PROVISIONAL)	Mental Health and Mental Wellbeing of Black Students at UK Universities: A Review and Thematic Synthesis
AUTHORS	Stoll, Nkasi; Yalipende, Yannick; Byrom, Nicola; HATCH, STEPHANI; Lempp, Heidi

VERSION 1 – REVIEW

REVIEWER	Nathalie Maltais Université du Québec à Rimouski, Nursing, pediatrics, psychiatry
REVIEW RETURNED	07-Jun-2021

GENERAL COMMENTS	In the abstract, it would be interesting to know more about the studies that were chosen for the literature review, Is there a model to support the design? In the Prisma flow chart we don't know if it was just l'abstract that was screened or is it all the 369 articles that were read? and what were the reasons for the exclusion... It is hard to reproduce this study if we don't know these things. I see now that in the article you are writing about it in details... good! I would have mentionned in the abstract Braun and Clarke's (2006)27 six-step guide. The method to obtain the data is well explained in the text!!! interesting that you have your own system to asses the studies. Very informative and educative
--

REVIEWER	R. Buckley Foothills Prov Gen Hosp
REVIEW RETURNED	07-Jun-2021

GENERAL COMMENTS	Good paper. Just a very few things to check on. 1) Table 3 - Reference Saran S. 2019 - had no "Xs" in the boxes. Is this correct info? It is different from the rest of the papers. Results - Line 29 - should be "waiting" and not "wating". Discussion - Line 5 - Please supply a reference for death of George Floyd. - Line 17 should be "students" and not "student".
---

REVIEWER	Javier Santabarbara University of Zaragoza Faculty of Medicine, Preventive Medicine
REVIEW RETURNED	17-Jun-2021

GENERAL COMMENTS	Thank you for the opportunity to review the manuscript entitled "Mental Health and Mental Wellbeing of Black Students at UK Universities: A Review and Thematic Synthesis" in BMJ Open.
---

	It is a good review and well written. This work represents an important area of inquiry that is relevant to the readership of this journal. However, additional details are needed for optimal review. 1. My main concern is related to the use of gray literature in a review (doctoral theses, dissertations, Open Gray ...). Specifically, 5 of the 12 selected articles come from this type of gray literature, which could affect the methodological quality of the review. This is a very important limitation that prompts a deep discussion. 2. Why do the authors report the PRISMA checklist when they have not provided a "systematic" review? It is not referenced in the text of the manuscript.
--	--

REVIEWER	Lisa Merlo University of Florida, Psychiatry
REVIEW RETURNED	18-Jun-2021

GENERAL COMMENTS	Thank you for the opportunity to review "Mental Health and Mental Wellbeing of Black Students at UK Universities: A Review and Thematic Synthesis." This is a timely and important topic of study, and a novel approach to examining the literature (i.e., review and thematic synthesis of qualitative studies on the topic). Some suggestions are offered below for consideration. Introduction: The background section provides a good primer for the reader, though it may be helpful to at least mention some information about mental health struggles for international students, as this is a significant focus of the Results. There are some confusing phrases/terms that would benefit from clarification. For example:  • Pg. 4 line 49-51, what does it mean that Black students are "more likely than White students to engage and participate in their university studies"? • Line 53, What does it mean to "achieve a good degree (1st or 2:1)"? When describing the aim of the review (pg. 5), it would be helpful to clarify that only qualitative research sources were included. Methods: The first sentence of the study design section indicates that quantitative studies were reviewed, but the second states that all included articles were qualitative, and there are no quantitative studies included in the Results. Please clarify. Results: Would it be possible to include the superscript numerical citations in the table so that the reader can more easily refer to them when reading the Results? When reviewing the themes, it would be helpful to use vocabulary that clarifies which results refer to students' beliefs/perceptions vs. which refer to more objective observation/assessment. The themes seem to be somewhat indistinct, with quite a bit of overlap in their descriptions and some repetition of findings across multiple themes. It may be useful to begin each new theme paragraph with an operational definition of the theme to ensure that the findings discussed in that section actually fit there. This may also help to better organize each theme subsection to provide a more cohesive summary/synthesis of the findings (some of the subsections are a bit disjointed as currently presented). There appear to be some assumptions in the results that extend beyond the actual findings of the studies (or need to be clarified). For example, on page 11 (line 23), it states that "Black students reported that unlike their non-Black peers, they faced pressures from their families" (italics added). Is there evidence that the non-Black
---

	students didn't face familial pressure? Please clarify. Then, later in the same paragraph, it says that "Black students with pre-existing mental and physical health conditions might be especially vulnerable..." The authors' ideas about what "might" be true would be more appropriately situated in the Discussion, as the Results section should focus on what the findings actually show. In the Isolation section, it is unclear whether the authors mean that the Black students struggled with their mental health before attending university, or whether they experienced stigma before attending university, or both. Was the stigma related to their mental health struggles? Were they isolated before university, during university, or both? Please clarify. Does the "Culture Shock" theme apply only to international students, or did some of the students from the UK also experience culture shock when leaving home to attend university? On page 14, what does it mean to say, "Black students' recognition of racial difference had a greater impact on their mental wellbeing and potential or experienced mental health difficulties, than evidence that others treated them differently on account of their race"? Does this mean that their perception of racism impacted them more than their objective experience of racism? How was their objective experience assessed? The first paragraph in the "Support" section would benefit from rewriting for clarity. On page 15, lines 28-30, was there evidence that Black students waited longer to seek help or faced longer waiting times than non-Black students for mental health services, or was their experience similar to other students? Discussion: The authors do a nice job relating their findings to the historical context and existing literature; however, the writing is a bit disjointed. It would be helpful to reorganize a bit to better synthesize the information into a more cohesive presentation to highlight the importance of the study findings. Other: there are a lot of missing words and other minor grammatical errors throughout the manuscript. Please read carefully and edit accordingly.
--	--

VERSION 1 – AUTHOR RESPONSE

Reviewer: 1

Prof. Nathalie Maltais, Université du Québec à Rimouski

Comments to the Author:

In the abstract, it would be interesting to know more about the studies that were chosen for the literature review, Is there a model to support the design? In the Prisma flow chart we don't know if it was just the abstract that was screened or is it all the 369 articles that were read? and what were the reasons for the exclusion... It is hard to reproduce this study if we don't know these things. I see now that in the article you are writing about it in details... good!

Thank you for your interest in the paper. We believe the reviewer was asking for clarification on the design and methods but then realised the details were in the manuscript, therefore no changes were recommended.

I would have mentioned in the abstract Braun and Clarke's (2006) 27 six-step guide. The method to obtain the data is well explained in the text!!! interesting that you have your own system to assess the studies. Very informative and educative

Thank you for this recommendation, reference to Braun and Clarke (2006) has been included in the abstract.

Reviewer: 2

Dr. R. Buckley, Foothills Prov Gen Hosp

Comments to the Author:

Good paper. Just a very few things to check on.

1) Table 3 - Reference Saran S. 2019 - had no "Xs" in the boxes. Is this correct info? It is different from the rest of the papers.

We are unsure of how to address this comment as the Bunce, L., King, N., Saran, S., & Talib, N. (2019) reference has X's in the boxes. It may be perhaps the reviewer missed this as the reference falls on pages 10 and 11, especially as the reviewer called the reference "Saran S. 2019", which is incorrect. Therefore, no changes have been made.

Results - Line 29 - should be "waiting" and not "wating".

Thank you for noticing this, the spelling correction has been rectified.

Discussion - Line 5 - Please supply a reference for death of George Floyd.

A reference to the mental health impact of the murder of George Floyd has been added to the manuscript which we hope further helps to ground the findings in the historical context.

- Line 17 should be "students" and not "student".

Thank you for noticing this, this spelling correction has been rectified.

Reviewer: 3

Dr. Javier Santabarbara, University of Zaragoza Faculty of Medicine

Comments to the Author:

Thank you for the opportunity to review the manuscript entitled "Mental Health and Mental Wellbeing of Black Students at UK Universities: A Review and Thematic Synthesis" in BMJ Open.

It is a good review and well written. This work represents an important area of inquiry that is relevant to the readership of this journal. However, additional details are needed for optimal review.

1. My main concern is related to the use of gray literature in a review (doctoral theses, dissertations, Open Gray ...). Specifically, 5 of the 12 selected articles come from this type of gray literature, which could affect the methodological quality of the review. This is a very important limitation that prompts a deep discussion.

The Critical Appraisal Skills Programme (CASP) Checklist was used to systematically assess the trustworthiness, relevance and results of articles, including grey literature. In the 'Strengths and Limitations' section we indicate grey literature as a strength due to the paper being about summarising the personal and institutional factors that influence the mental health of Black university students (see page 18) The doctoral theses and dissertations included in our paper focused on these factors which adds value to the review, rather than

limitations. However, we are aware there are wider debates in the literature about the strengths and limitations of grey literature.

2. Why do the authors report the PRISMA checklist when they have not provided a "systematic" review? It is not referenced in the text of the manuscript.

The PRISMA checklist included in the "Research Checklist" has been removed from the files as suggested.

Reviewer: 4

Dr. Lisa Merlo, University of Florida

Comments to the Author:

Thank you for the opportunity to review "Mental Health and Mental Wellbeing of Black Students at UK Universities: A Review and Thematic Synthesis." This is a timely and important topic of study, and a novel approach to examining the literature (i.e., review and thematic synthesis of qualitative studies on the topic). Some suggestions are offered below for consideration.

Introduction: The background section provides a good primer for the reader, though it may be helpful to at least mention some information about mental health struggles for international students, as this is a significant focus of the Results.

Thank you for this feedback. The theme "culture shock" (on page13) is the only section of the results that discusses the experiences of international students and only one paper was found to focus specifically on international students. We have added clarification to the references in the introduction that included British and international students (page 4); however, the only relevant study that we could include is the one we found during the search.

There are some confusing phrases/terms that would benefit from clarification. For example:

- Pg. 4 line 49-51, what does it mean that Black students are "more likely than White students to engage and participate in their university studies"?
- Line 53, What does it mean to "achieve a good degree (1st or 2:1)"?

Thank you for this feedback, we have amended the manuscript to be more specific and detailed about what the authors meant by "more likely than White students to engage and participate in their university studies" (page 5). A "good degree (1st or 2:1)" (page 5) is a UK specific term based on the degree classification system. As this paper is UK specific we believe no changes need to be made to this statement. We have carefully reviewed and edited potential confusing phrases/terms in the manuscript. We have also reviewed and edited the manuscript to make sure all phrases and terms are clear.

When describing the aim of the review (pg. 5), it would be helpful to clarify that only qualitative research sources were included.

Thank you for this suggestion but the aim of the review was not to limit the review to qualitative studies, but once the search was completed, only qualitative studies met the inclusion criteria. We have discussed this limitation in the 'Strengths and Limitations' section on page 18 of the manuscript.

Methods: The first sentence of the study design section indicates that quantitative studies were

reviewed, but the second states that all included articles were qualitative, and there are no quantitative studies included in the Results. Please clarify.

Similar to the above comment, in the manuscript we state that the aim of the review was to review quantitative, qualitative and mixed studies, but once the search was complete only qualitative studies met the inclusion criteria. In the discussion we mention this as a limitation on page 18.

Results: Would it be possible to include the superscript numerical citations in the table so that the reader can more easily refer to them when reading the Results?

Thank you for this helpful suggestion, the references have been added to all three tables.

When reviewing the themes, it would be helpful to use vocabulary that clarifies which results refer to students' beliefs/perceptions vs. which refer to more objective observation/assessment. The themes seem to be somewhat indistinct, with quite a bit of overlap in their descriptions and some repetition of findings across multiple themes. It may be useful to begin each new theme paragraph with an operational definition of the theme to ensure that the findings discussed in that section actually fit there. This may also help to better organize each theme subsection to provide a more cohesive summary/synthesis of the findings (some of the subsections are a bit disjointed as currently presented). There appear to be some assumptions in the results that extend beyond the actual findings of the studies (or need to be clarified). For example, on page 11 (line 23), it states that "Black students reported that unlike their non-Black peers, they faced pressures from their families" (italics added). Is there evidence that the non-Black students didn't face familial pressure? Please clarify. Then, later in the same paragraph, it says that "Black students with pre-existing mental and physical health conditions might be especially vulnerable..." The authors' ideas about what "might" be true would be more appropriately situated in the Discussion, as the Results section should focus on what the findings actually show.

Thank you for this very helpful feedback. The authors have revised the manuscript to include an opening sentence for each theme and made it clear we are synthesising students' beliefs and experiences, not objective observations or assessments. As this is a qualitative review of Black students' experiences, none of the findings are objective. The authors have added a couple of sentences under the heading 'Synthesised findings' on page 10 to clarify.

In the Isolation section, it is unclear whether the authors mean that the Black students struggled with their mental health before attending university, or whether they experienced stigma before attending university, or both. Was the stigma related to their mental health struggles? Were they isolated before university, during university, or both? Please clarify.

In the manuscript we refer to isolation and stigma affecting Black students' mental health both before and during university. This is indicated in the two sentences written in the isolation theme on page 12 and 13, one referring to "before attending university" and one sentence referring to "at university".

Does the "Culture Shock" theme apply only to international students, or did some of the students from the UK also experience culture shock when leaving home to attend university?

There was no evidence from included studies to illustrate culture shock from British students; but we recognise that the impact of culture shock on Black British students' mental health

warrants future investigation. In the discussion we recommend further research into the impact migration has on Black student mental health (page 16).

On page 14, what does it mean to say, “Black students’ recognition of racial difference had a greater impact on their mental wellbeing and potential or experienced mental health difficulties, than evidence that others treated them differently on account of their race”? Does this mean that their perception of racism impacted them more than their objective experience of racism? How was their objective experience assessed?

The articles from which this statement was based upon refers to expected racism, however there was no exploration into the assessment of perceived, expected, experienced racism. This requires further investigation, which is not in the scope of this study. However, in the discussion we recommend further research into the role anti-Black racism plays on mental health inequality.

The first paragraph in the “Support” section would benefit from rewriting for clarity.

Thank you for the helpful feedback, we agree and have rewritten the first paragraph of the support theme on page 14 and 15 and believe it reads much better now.

On page 15, lines 28-30, was there evidence that Black students waited longer to seek help or faced longer waiting times than non-Black students for mental health services, or was their experience similar to other students?

There was no evidence that Black students waited longer, as indicated by the statement that “when they did, faced long waiting times”, not longer, in the manuscript.

Discussion: The authors do a nice job relating their findings to the historical context and existing literature; however, the writing is a bit disjointed. It would be helpful to reorganize a bit to better synthesize the information into a more cohesive presentation to highlight the importance of the study findings.

Thank you for the helpful feedback, we agree and have restructured and rewritten the discussion to integrate the study findings with the historical context and existing literature more cohesively. We are hopeful the discussion reads much better now.

Other: there are a lot of missing words and other minor grammatical errors throughout the manuscript. Please read carefully and edit accordingly.

Thank you for the helpful feedback and close attention paid to the manuscript. We have edited the manuscript to correct missing words and grammatical errors.

VERSION 2 – REVIEW

REVIEWER	Javier Santabarbara University of Zaragoza Faculty of Medicine, Preventive Medicine
REVIEW RETURNED	17-Oct-2021
GENERAL COMMENTS	The authors improved the manuscript

REVIEWER	Lisa Merlo University of Florida, Psychiatry
REVIEW RETURNED	26-Oct-2021

GENERAL COMMENTS	Thank you for the opportunity to re-review “Mental health and mental wellbeing of Black students at UK universities: A review and thematic synthesis.” Overall, this is an interesting paper that makes an important contribution to the literature on Black university student wellbeing in the UK—an understudied but critical topic. The authors were generally responsive to reviewer recommendations, and the current draft is much improved. There are some remaining issues with grammar/awkward phrasing throughout, so the manuscript would benefit from careful proof-reading. Other minor suggestions are included below. Page 5, line 23—it would be helpful to clarify that you are referring to the “university student lifestyle” that was previously described. Page 6, line 5—it is unclear to international readers what it means to “achieve a good degree (1st or 2:1).” Page 6, line 19—It may be preferable to say “this review aims to synthesize results of existing studies.” Page 16, line 17—Do you mean that families were “continuously reminding them of the sacrifices and investments made” to support their education? Page 16, line 24—do you mean they feared “a mental illness diagnosis might affect success? Or was there objective/observable evidence that it would? Page 16, line 40—as written, it is unclear whether you mean Black students felt they were racially stereotyped/reported believing they were racially stereotyped, or that there was objective/observable evidence of such. (Note: I am not questioning whether the students were stereotyped, and I am aware—from the response to reviewers—that the data reflect student perceptions. However, this needs to be clarified in the actual text. Page 16, line 43-48—this is confusing as written. Please edit for clarity. Page 17, line 31—what does it mean that the students “experienced stigma from the Black community”? Can you provide some specific examples or description? Page 18, line 35—Do you mean that students experienced increased stress in their personal lives or due to external (i.e., sociopolitical) issues? Or both? Page 19, line 45—the word “health” is missing in “mental health and wellbeing”
--

VERSION 2 – AUTHOR RESPONSE

Reviewer: 3

Dr. Javier Santabarbara, University of Zaragoza Faculty of Medicine

Comments to the Author:

The authors improved the manuscript

Thank you for reviewing our manuscript and we are happy to see our changes have been received well.

Reviewer: 4

Dr. Lisa Merlo, University of Florida

Comments to the Author:

Thank you for the opportunity to re-review “Mental health and mental wellbeing of Black students at UK universities: A review and thematic synthesis.” Overall, this is an interesting paper that makes an important contribution to the literature on Black university student wellbeing in the UK—an understudied but critical topic. The authors were generally responsive to reviewer recommendations, and the current draft is much improved. There are some remaining issues with grammar/awkward phrasing throughout, so the manuscript would benefit from careful proof-reading. Other minor suggestions are included below.

Thank you for re-reviewing our manuscript. As advised, we have done careful proofreading of the manuscript and corrected the spelling and grammar, and made small changes to sentence structuring throughout the manuscript. We hope the manuscript reads more clearly now and appreciate the time you have taken to carefully review and comment on our work.

Page 5, line 23—it would be helpful to clarify that you are referring to the “university student lifestyle” that was previously described.

Thank you for this recommendation, we have included the reference every time the university student lifecycle is mentioned, for clarity (page 4).

Page 6, line 5—it is unclear to international readers what it means to “achieve a good degree (1st or 2:1).”

We have written out the classification, included the percentage, and made it clear that it is the UK degree classification system. We think the new insertions will be clearer to international readers (page 5).

Page 6, line 19—It may be preferable to say “this review aims to synthesize results of existing studies.”

Thank you for this correction, we have made the amendment following your suggestion (page 5).

Page 16, line 17—Do you mean that families were “continuously reminding them of the sacrifices and investments made” to support their education?

Yes we did, thank you for the suggestion, we have made the amendment and the sentence reads better now (page 11).

Page 16, line 24—do you mean they feared “a mental illness diagnosis might affect success? Or was there objective/observable evidence that it would?

Yes we did, we have made the amendment to make this clearer (page 11).

Page 16, line 40—as written, it is unclear whether you mean Black students felt they were racially stereotyped/reported believing they were racially stereotyped, or that there was objective/observable evidence of such. (Note: I am not questioning whether the students were stereotyped, and I am aware—from the response to reviewers—that the data reflect student perceptions. However, this needs to be clarified in the actual text.

Thank you for this comment, we agree, and have reworded the sentence to make it clear that the students recalled racially stereotypes being shared in seminars and reinforced in course materials (page 12).

Page 16, line 43-48—this is confusing as written. Please edit for clarity.

We have reworded this sentence and think it is easier to read (page 12).

Page 17, line 31—what does it mean that the students “experienced stigma from the Black community”? Can you provide some specific examples or description?

Thank you for this comment, we have decided to re-word this sentence to remove the generalisation of “the Black community” to “their community” and clarified the community is made of friends and family as the latter is more grounded in the participants’ reports (page 13).

Page 18, line 35—Do you mean that students experienced increased stress in their personal lives or due to external (i.e., sociopolitical) issues? Or both?

We have provided more context and changed the wording. We believe the amendment provides more clarification that the stress were personal issues, with no mention of socio-political issues in the paper (page 14).

Page 19, line 45—the word “health” is missing in “mental health and wellbeing”

Thank you for catching this, we have now amended (page 11).